# Structural dynamics of the CROPs domain control stability and toxicity of *Paeniclostridium sordellii* lethal toxin

Yao Zhou[1,2,3,4,5], Xiechao Zhan ✉[1,2,3,5], Jianhua Luo[1,2,3,4], Diyin Li[1,2,3,4], Ruoyu Zhou[1,2,3], Jiahao Zhang[1,2,3], Zhenrui Pan[1,2,3], Yuanyuan Zhang[1,2,3], Tianhui Jia[1,2,3,4], Xiaofeng Zhang[1,2,3], Yanyan Li[1,2,3] & Liang Tao ✉[1,2,3,4]

*Paeniclostridium sordellii* lethal toxin (TcsL) is a potent exotoxin that causes lethal toxic shock syndrome associated with fulminant bacterial infections. TcsL belongs to the large clostridial toxin (LCT) family. Here, we report that TcsL with varied lengths of combined repetitive oligopeptides (CROPs) deleted show increased autoproteolysis as well as higher cytotoxicity. We next present cryo-EM structures of full-length TcsL, at neutral (pH 7.4) and acidic (pH 5.0) conditions. The TcsL at neutral pH exhibits in the open conformation, which resembles reported TcdB structures. Low pH induces the conformational change of partial TcsL to the closed form. Two intracellular interfaces are observed in the closed conformation, which possibly locks the cysteine protease domain and hinders the binding of the host receptor. Our findings provide insights into the structure and function of TcsL and reveal mechanisms for CROPs-mediated modulation of autoproteolysis and cytotoxicity, which could be common across the LCT family.

*P aeniclostridium sordellii* (also known as *Clostridium sordellii*) is a gram-positive anaerobic bacterium commonly found in the environment and causes fatal infections in animals like cattle, sheep, and horses[1]. In humans, *P. sordellii* infections are not often but very severe, leading to life-threatening edema, gangrene, hypotension, and sepsis with overall mortality rates approaching ~70%[2,3]. Approximately 3-4% of women in the United States carry *P. sordellii* in the vaginal and rectal tract; this number is further increased if a recent gynecologic procedure exists[4]. Clinical reports showed that during gynecological procedures, such as childbirth, miscarriage, and abortion, women were particularly at high risk for fulminant *P. sordellii* infections with a 100% mortality[5–8].

*P. sordellii* expresses lethal toxin TcsL (~270 kDa) and hemorrhagic toxin TcsH (~300 kDa). TcsL can cause major damage to endothelial and epithelial cells, particularly in the lungs, and is thought to be the primary cause of the high mortality for acute *P.*

*sordellii* infections[9,10]. Both TcsL and TcsH belong to the LCT family, which also includes TcdA and TcdB in *Clostridioides difficile*, TpeL in *Clostridium perfringens*, and Tcnα in *Clostridium novyi*. Among LCTs, TcsL is most closely related to TcdB, with a sequence identity of ~75%[11]. LCT family members, including TcsL consist of four functional domains: an N-terminal glucosyltransferase domain (GTD), a cysteine protease domain (CPD) that mediates the autocleavage, an intermingled domain responsible for both transmembrane delivery and receptor-binding (DRBD), and a C-terminal combined repetitive oligopeptides (CROPs) domain[12,13]. Once the toxins enter target cells via receptor-mediated endocytosis, their GTD and CPD translocate from the lumen of the endosomes to the outside upon low pH[14–16]. In the cytosol, the release of the GTD depends on the protease activity of the CPD triggered by the binding of inositol-hexakisphosphate (InsP6)[17]. The activation of CPD results in autocleavage between Leu543 and Gly544 in TcdB and TcsL[18–20]. The released GTD then

[1]Center for Infectious Disease Research, Westlake Laboratory of Life Sciences and Biomedicine, Hangzhou, Zhejiang 310024, China. [2]Key Laboratory of Structural Biology of Zhejiang Province, School of Life Sciences, Westlake University, Hangzhou, Zhejiang 310024, China. [3]Westlake Institute for Advanced Study, Hangzhou, Zhejiang 310024, China. [4]Research Center for Industries of the Future, Westlake University, Hangzhou, Zhejiang 310024, China. [5]These authors contributed equally: Yao Zhou, Xiechao Zhan. ✉e-mail: zhanxiechao@westlake.edu.cn; taoliang@westlake.edu.cn

glucosylates small GTPase proteins, leading to actin cytoskeleton disruption and cell death[21–26].

Great efforts have been made to understand the toxin action mechanisms of the LCTs through structural biological approaches[21,27]. Recently, some high-resolution full-length or near-complete structures for TcdA[28–30] and TcdB[31–33] were reported. In those structures, the "core" domains (GTD, CPD, and DRBD) show a similar overall architecture, but the CROPs seem to be dynamic and have at least two major conformations: one curves upward around the GTD-CPD head and is referred to as the "open" conformation, the other curves downward alongside the DRBD and is referred to as the "closed" conformation. TcdA is usually displayed in the closed conformation[29,30,34]. TcdB tends to display the open conformation in either the crystal or cryogenic electron microscopy (cryo-EM) structures[31–33] while its closed form has been detected using SAX or XL-MS analyses[31,34]. In previous studies, small regions located at the N-terminal boundary of TcdA and TcdB CROPs were defined as the "hinge", which may be responsible for the swing of the CROPs domains[29,31]. To our knowledge, no full-length structures for other LCTs have ever been reported.

The CROPs are unique structural modules found in almost all LCTs except for TpeL. These domains are composed of multiple repeating units: 19- to 24-amino acid short repeats (SRs) interspersed with 29- to 31-amino acid long repeats (LRs)[27]. The LCT CROPs are variable in sequence and length, ranging from ~350 to ~900 amino acids, while TpeL is a natural CROP-less LCT[35]. For TcsL, the CROPs domain is composed of four equivalent units (namely I–IV), each unit contains five SRs and one LR. The CROPs were thought to be the receptor-binding domains in LCTs mainly due to the early studies on TcdA. It was shown that the CROPs domain of TcdA was capable of binding the trisaccharide Gal-α1,3-Gal-β1,4-GlcNAc and thus considered as a domain responsible for binding carbohydrate moieties on cell surface[36,37]. In recent years, multiple

protein receptors for LCTs were identified, including low-density lipoprotein receptor-related protein 1 (LRP1) for TpeL[38], low-density lipoprotein receptor (LDLR) family proteins for TcdA and Tcnα[39–42], chondroitin sulfate proteoglycan 4 (CSPG4), poliovirus receptor-like 3 (PVRL3), Frizzled proteins (FZDs), and tissue factor pathway inhibitor (TFPI) for TcdB[33,43–46], Semaphorin 6 A and 6B (SEMA6A/6B) for TcsL[47,48], and transmembrane serine protease 2 (TMPRSS2) for TcsH[49]. Many of them were structurally validated to bind LCTs independent of the CROPs[33,48,50,51], raising the speculation that the CROPs in LCTs are not solely for host recognition but have uncharacterized functions.

Here, we found that the CROPs completely or partially removed TcsL had remarkably increased autoproteolysis as well as cytotoxicity compared to the full-length toxin. To investigate the molecular basis of such CROPs-dependent regulations, we further determined the architecture of full-length TcsL at both neutral and acidic conditions by cryo-EM and captured two major conformations (open and closed). Both functional and structural analysis revealed two regions in the TcsL CROPs that are critical for the modulations: one closed to the DRBD (partly overlapped with the hinge region) and the other at the end of the CROPs domain. Lastly, we demonstrated that an integrated CROPs domain is important to retain the in vivo toxicity of TcsL pre-exposed to InsP6.

## Results

### The CROPs robustly inhibit the autoproteolytic activity of CPD in TcsL

To investigate the potential roles of the CROPs domain in TcsL, we set out to generate the full-length TcsL and a TcsL truncation (TcsL[1–1808]) with its entire CROPs deleted (Fig. 1a). These toxin proteins (and other LCTs thereafter) were C-terminal His-tagged and expressed in *Bacillus subtilis*. Unexpectedly, we found that TcsL[1–1808] but not full-length TcsL

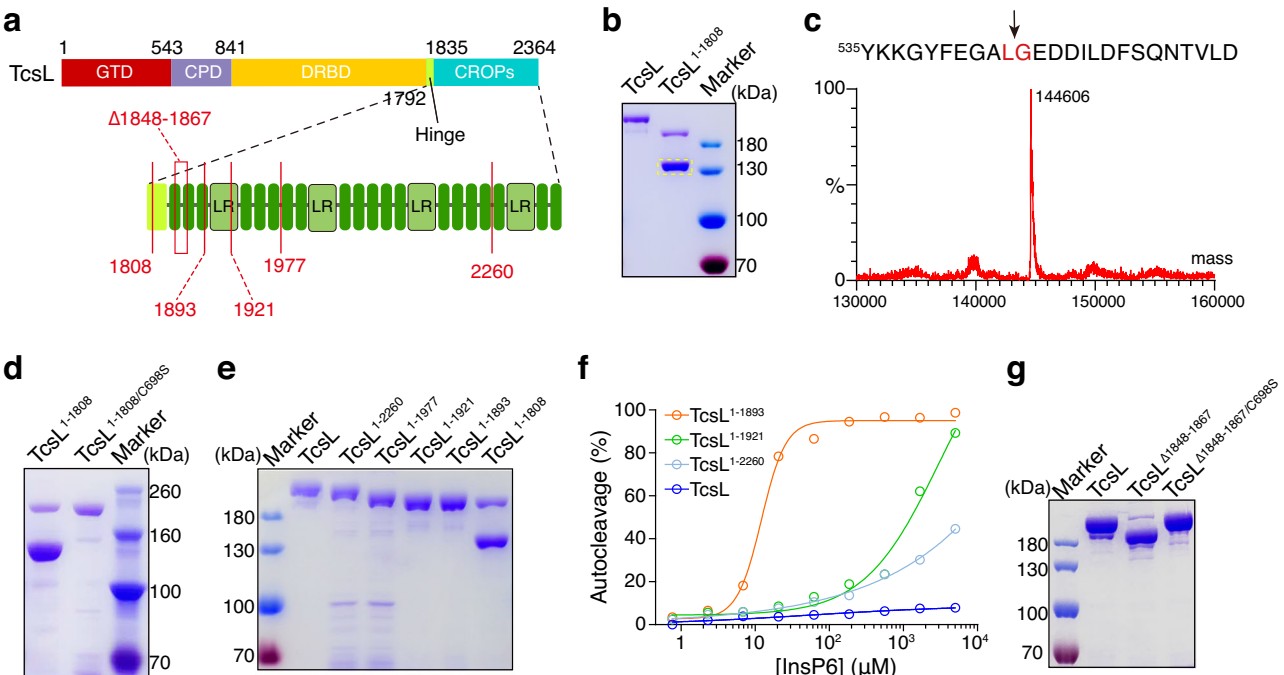

**Fig. 1 | The CROPs domain protects TcsL from autoproteolysis. a** A Schematic drawing of the TcsL truncations designed for this study. **b** Purified full-length TcsL and TcsL[1–1808] were separated on a Coomassie-stained SDS-PAGE gel. TcsL[1–1808] was largely fractured, leaving a ~140 kDa fragment (light dotted box). **c** The large fragment in (b) was analyzed by mass spectrometry. The black arrows indicate a breaking point between Leu543 and Gly544. **d** Purified TcsL[1–1808] and TcsL[1–1808/C698S] were separated on a Coomassie-stained SDS-PAGE gel. **e** Purified full-length TcsL, TcsL[1–2260],

TcsL[1–1977], TcsL[1–1921], TcsL[1–1893], and TcsL[1–1808] were separated on a coomassie-stained SDS-PAGE gel, only TcsL[1–1808] showed overt autoproteolysis. **f** The autocleavage of TcsL, TcsL[1–2260], TcsL[1–1921], and TcsL[1–1893] was induced by the gradient concentrations of InsP6 for 3 hours. The percentage of cleavage was measured and plotted on the chart. **g** Purified full-length TcsL, TcsL[Δ1848-1869], and TcsL[Δ1848-1869/C698S] were separated on a coomassie-stained SDS-PAGE gel. Source data are provided as a Source Data file.

was largely fractured after the purification, leaving a ~140 kDa C-terminal fragment (Fig. 1b).

Mass spectral (MS) analysis of the ~140 kDa fragment showed that the breakage happened between Leu543 and Gly544 (Fig. 1c), which is the reported cleavage site for CPD[18,20]. We then mutated Cysteine at position 698 to Serine, which abolishes the proteolytic activity of the CPD[52,53], in TcsL$^{1-1808}$. Unlike TcsL$^{1-1808}$, TcsL$^{1-1808/C698S}$ was purified as an intact polypeptide (Fig. 1d), demonstrating that the breakage in TcsL$^{1-1808}$ is due to excessive autocleavage by its own CPD.

## The CROPs suppress the autoproteolysis of TcsL in two manners

To determine how much the CROPs were required to prohibit the autoproteolysis of TcsL, we next generated several C-terminally truncated TcsL with partial CROPs remained, including TcsL$^{1-2260}$, TcsL$^{1-1977}$, TcsL$^{1-1921}$, and TcsL$^{1-1893}$. None of them showed obvious autoproteolysis, which is similar to the full-length TcsL but not TcsL$^{1-1808}$ (Fig. 1e, f), indicating that the region between residues 1809 and 1893 tightly controls the autoproteolytic activity of the CPD. In addition, a TcsL mutant with 19 residues (residues 1848 to 1867) deleted also had substantial CPD-mediated autocleavage (Fig. 1g), further supporting that this small region is critical for inhibiting the autoproteolysis in TcsL.

We then quantitatively measured the rates of autoproteolysis for TcsL, TcsL$^{1-2260}$, TcsL$^{1-1921}$, and TcsL$^{1-1893}$ in the presence of exogenous InsP6. The full-length TcsL is resistant to InsP6-induced autoproteolysis, which is consistent with a previous report[54]. Although these truncations are generally stable upon production, they showed varied autoprocessing rates when exposed to gradient concentrations of InsP6: TcsL derivates with shorter CROPs remaining could be induced by InsP6 more easily (Fig. 1f, Supplementary Fig. 1).

## The C-terminus of CROPs affects the cytotoxicity of TcsL

To examine the cytotoxicity of these TcsL truncates, we performed the cytopathic cell-rounding assays on HeLa and A549 cells. To our surprise, the C-terminal truncated TcsL, including TcsL$^{1-2260}$, TcsL$^{1-1977}$, TcsL$^{1-1921}$, and TcsL$^{1-1893}$, were more potent than full-length TcsL in the tested cells (Fig. 2a). Albeit A549 cells are more susceptible to TcsL due to the differed expression of SEMA6A/6B[47,48], a similar pattern was observed in both HeLa and A549 cells: the truncated toxins showed close toxicity to each other and had ~30-fold increased cytotoxicity compared to full-length TcsL (Fig. 2b). Previous studies reported that CROPs-truncated TcdA, TcdB, and TcsH usually had similar or reduced (in varying degrees) cytotoxicity compared to the full-length toxins, due to the weakened binding to target cells[42,45,49,55,56].

Since TcsL is sequentially close to TcdB, including the CROPs (Supplementary Fig. 2), we then replaced the last 104 amino acids (residues 2261 to 2364) with the homologous sequence from TcdB (more precisely, TcdB1[11]). This newly built chimeric toxin was named TcsL$^{2261B}$ (Fig. 2c). TcsL$^{2261B}$ also showed similar cytotoxicity to TcsL truncates and was more potent than the full-length TcsL (Fig. 2d). In addition, TcsL$^{2261B}$ showed stronger InsP6-induced autocleavage (Fig. 2e, Supplementary Fig. 3), which resembled the TcsL truncates but not TcsL. These results indicate that the TcsL CROPs unit-IV can specifically suppress the cytotoxicity and InsP6-induced autoproteolysis of TcsL with sequence specificity to a certain extent.

## Cryo-EM structure of the full-length TcsL

To characterize the potential molecular mechanisms of CROPs-mediated modulations on autoproteolysis and cytotoxicity of TcsL, we managed to determine the structure of the full-length TcsL at neutral pH using single particle cryo-EM. The two-dimensional class averages of TcsL at neutral pH exhibited clear structural features, and the final reconstruction of the full-length TcsL generated an EM map at 2.9 Å resolution (Supplementary Fig. 4).

The cryo-EM structure of TcsL reveals the four major domains of the toxin as previously described (Fig. 3a). The GTD and CPD domains constitute the central core modules of the toxin, interacting with the DRBD and CROPs domains. The DRBD extended and pointed away from the GTD and CPD. The CROPs domain emerges from the junction and curves upward around the core region like a hook, which forms a typical open conformation (Fig. 3a). The overall architecture of TcsL closely resembled that of previously reported structures of TcdB variants as determined by either crystallization or cryo-EM[31–33].

## Low pH induces conformation change of TcsL

We next looked at the configuration of the full-length TcsL in a lowered pH (pH 5.0) using the single particle cryo-EM. Interestingly, we observed that TcsL at pH 5.0 simultaneously exhibits both open (~71.8%) and closed (~28.2%) conformations (Fig. 3b, Supplementary Fig. 5). The EM maps of the full-length TcsL at pH 5.0 were resolved at 2.9 Å (closed conformation) and 2.5 Å (open conformation) (Supplementary Fig. 6). The open conformation is close to the one obtained at neutral pH with a root-mean-square-deviation (RMSD) of 1.4 Å over 1856 Cα atoms (Fig. 3c, Supplementary Fig. 7a). In the closed conformation, the CROPs extend from the hinge region, curve around the section of the DRBD, and interact with the "tip" of the DRBD (Fig. 3d).

When superimposed the cryo-EM structures of TcsL at acidic pH in the open and closed conformations, with an RMSD of 5.1 Å over 1728 Cα atoms, the core domains, including GTD, CPD, and DRBD, are relatively identical, but a near-180° rotation of the entire CROPs domain were observed (Fig. 4a, Supplementary Fig. 7b). A major allosteric transition upon low pH induction was observed for the hinge region (residues 1792-1834) and the first SR swing for ~60 Å. As a result, the entire CROPs rotated for ~180° and the CROPs unit-IV touched the "tip" part of the DRBD (Fig. 4b, Supplementary Movie 1).

In the closed-form structure of TcsL, the CROPs domain potentially interacts with the DRBD through two intra-molecular interfaces (Fig. 4b, c). Interface I is formed between the Hinge region and the DRBD, while interface II is formed between the CROPs unit-IV and the middle part (residues 1176-1277) of the DRBD (Fig. 4b, c). Thus, low pH induces a conformational switch of TcsL from open to closed state, which is also in part supported by a previous XL-MS analysis for TcdB[31].

## CROPs retain the in vivo toxicity of TcsL upon exposure to InsP6

As an exotoxin, TcsL is naturally secreted by *P. sordellii* into its residing niches and exposed to environmental substances. Soils and feces, where *P. sordellii* normally resides, occasionally contain considerable amounts of InsP6[57]. We pre-exposed TcsL and TcsL$^{1-1808}$ with and without InsP6, followed by intraperitoneally (IP) injecting into mice. This procedure mimics the exposure of TcsL to extracellular InsP6 before entering the host cells. TcsL can induce toxic shock syndrome with similar vascular permeability and massive edema observed in lungs when intraperitoneally injected into mice[9,47,48]. Both TcsL and TcsL$^{1-1808}$ induced strong lung tissue damage at similar levels by evaluating both accumulated thoracic fluid (Fig. 5a, b) and hematoxylin and eosin (H&E) stained histological lung sections (Fig. 5c, d), suggesting that CROP-less TcsL is near-equally potent to TcsL once directly enter the host circulation system. Then we pre-mixed the toxins with InsP6 and then IP injected them into the mice. This serves as a simulation that the tested exotoxins are first exposed to environmental circumstances and then reach the host target. TcsL preincubated with or without InsP6 for 30 minutes induced similar thoracic fluid accumulation (Fig. 5a, b) and lung damage in mice (Fig. 5c, d), suggesting that the full-length TcsL is tolerant to environmental InsP6 to a certain extent. In contrast, TcsL$^{1-1808}$ preincubated with InsP6 for only five minutes showed drastically reduced capability in inducing tissue damage, while TcsL$^{1-1808}$ exposed to InsP6 for 30 minutes was nearly avirulent in vivo (Fig. 5a-d). These results demonstrated that

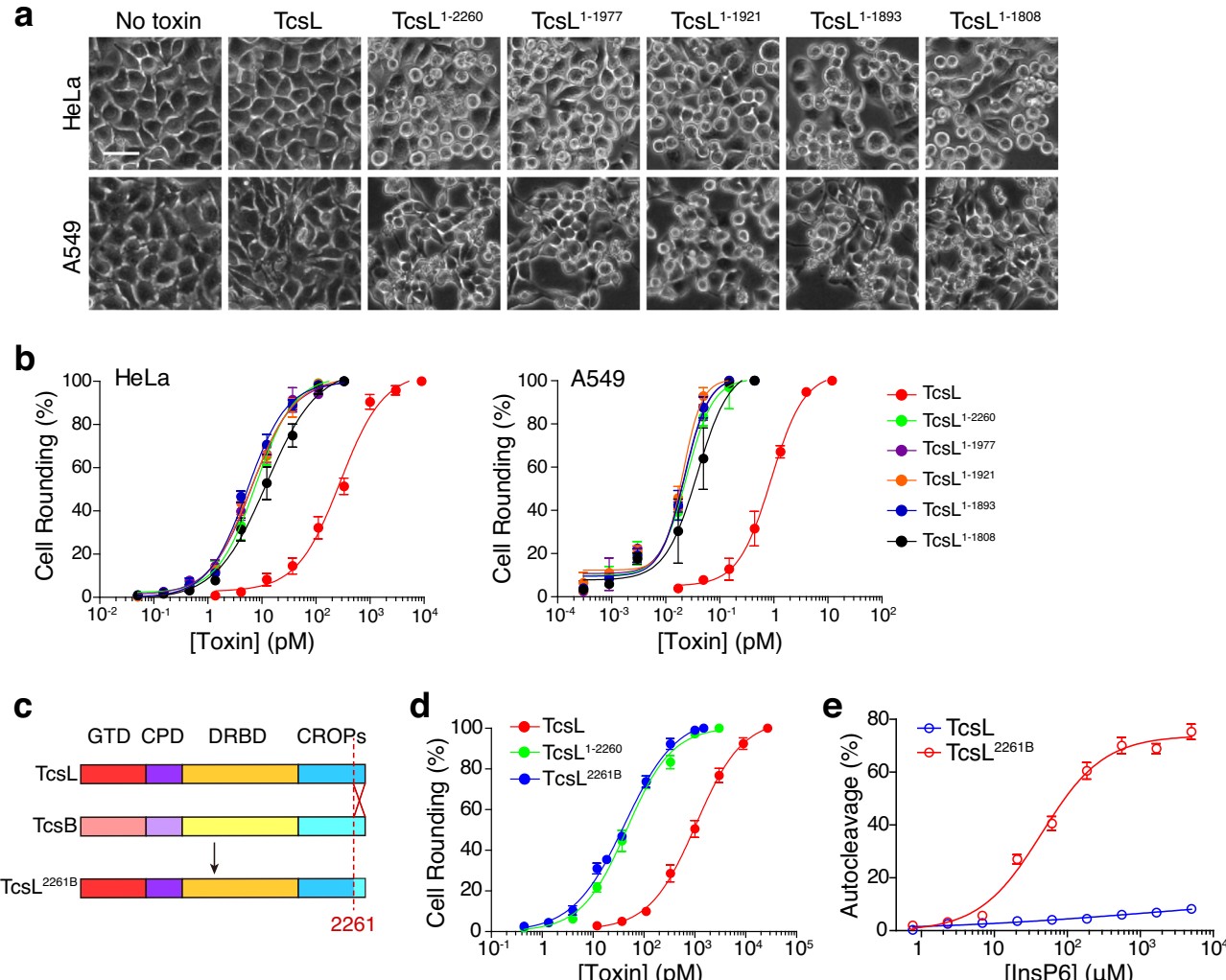

**Fig. 2 | The C-terminus of CROPs affects the cytotoxicity of TcsL. a** HeLa and A549 cells were incubated with TcsL, TcsL[1-2260], TcsL[1-1977], TcsL[1-1921], TcsL[1-1893], or TcsL[1-1808] (37 pM for HeLa, 0.05 pM for A549) for 24 hours. Representative images are shown. The scale bar represents 50 µm. **b** The sensitivities of HeLa and A549 cells to TcsL, TcsL[1-2260], TcsL[1-1977], TcsL[1-1921], TcsL[1-1893], TcsL[1-1808] were measured using cytopathic cell-rounding experiments. Error bars indicate mean ± SD, *n* = 6 biologically independent samples. **c** The schematic illustration of designed chimeric

toxin TcsL[2261B] based on TcsL and TcdB. **d** The sensitivities of HeLa cells to TcsL, TcsL[1-2260], and TcsL[2261B] were measured using cytopathic cell-rounding experiments. Error bars indicate mean ± SD, *n* = 6 biologically independent samples. **e** The autocleavage of TcsL[2261B] was induced by the gradient concentrations of InsP6 for 3 hours. The percentage of cleavage was measured and plotted on the chart. Error bars indicate mean ± SD, *n* = 3 independent experiments. Source data are provided as a Source Data file.

CROPs-dependent autoproteolysis inhibition could be functionally important for maintaining the toxicity of TcsL before host invasion.

## CROPs-mediated regulation on autoproteolysis is common for other LCTs

Next, we managed to investigate whether the CROPs-mediated inhibition on autoproteolysis is general across the LCT family. For this purpose, we produced the full-length toxins of other LCTs, including TcdA, TcdB, TcsH, and Tcnα, and their CROP-less versions (TcdA[1-1834], TcdB[1-1805], TcsH[1-1832], and Tcnα[1-1801]). These toxins were exposed to 100 µM InsP6 to test their autocleavage efficiency. Similar to TcsL, all tested LCTs in the full-length form showed weaker InsP6-induced autoproteolysis when compared to their CROP-less versions (Fig. 6a, Supplementary Fig. 8).

Finally, we simulated the situations in which TcdA and TcdB were pre-exposed to environmental InsP6 and then intoxicated the mice. TcdA, TcdA[1-1834], TcdB, and TcdB[1-1805] were pre-incubated with or without 100 µM InsP6 for 30 minutes and IP injected into the mice. InsP6-treated TcdA and TcdB could effectively kill the mice (Fig. 6b,

c), indicating that these full-length toxins are resistant to extracellular InsP6 with their toxicity largely retained. In contrast, InsP6-treated CROP-less TcdA and TcdB quickly lost their toxicity and became much less potent to mice (Fig. 6b, c), suggesting that the CROPs prevent the autoproteolysis of TcdA and TcdB induced by the extracellular InsP6.

## Discussion

As exotoxins, LCTs, including TcsL are secreted from the bacteria into their living niches and required to be structurally stable before reaching their target cells. Notably, InsP6 is sometimes distributed in various environments such as soil[58] and human gastrointestinal tract[59] where pathogenic clostridial species including *P. sordellii* and *C. difficile* colonize. To maintain the toxin function, LCTs need to prevent environmental factors (such as InsP6)-induced autoproteolysis and degradation. Our findings indicate that a brief five-minute pre-incubation of TcsL[1-1808] with InsP6 significantly reduced the majority of toxin activity in a mouse model. Conversely, the toxicity of full-length TcsL persisted unaltered 30 minutes after exposure to InsP6. Similar

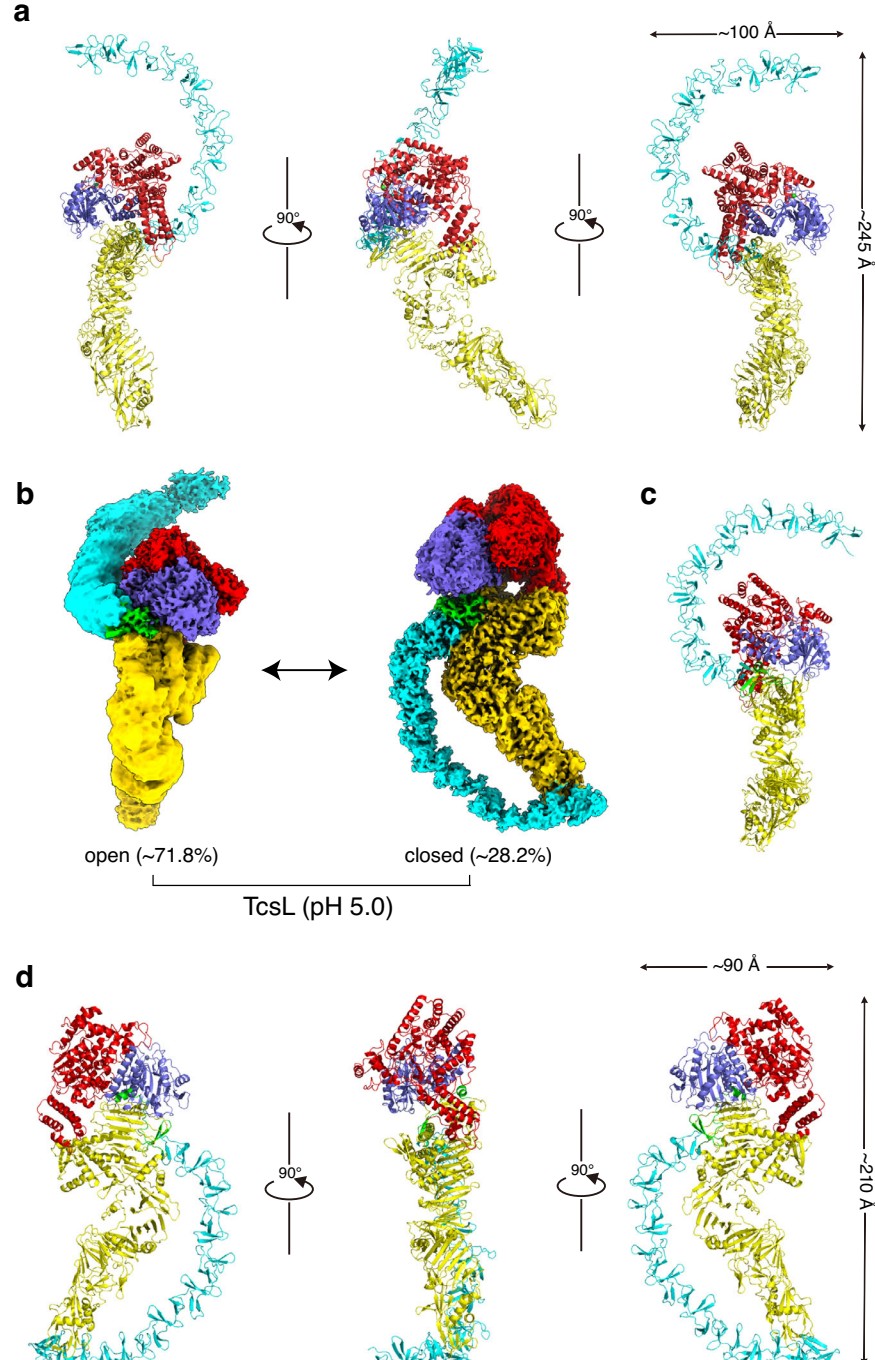

**Fig. 3 | Overall structures of the full-length TcsL. a** Cryo-EM structure of TcsL at neutral pH. **b** The EM-maps of TcsL at an acidic pH of 5.0 in both open (left) and closed (right) conformations. **c** Cryo-EM structure of TcsL in open conformation at pH 5.0. **d** Cryo-EM structure of TcsL in closed conformation at pH 5.0. The GTD, CPD, DRBD, and CROPs are shown in red, purple, yellow, and cyan, respectively. The zinc atom is shown as a green sphere.

results were obtained for other LCTs like TcdA and TcdB as well. Thus, there is a compelling necessity for LCTs, such as TcsL, to develop an extensive CROPs domain to stabilize the toxin. The only exception is TpeL, which is naturally CROP-less. How TpeL stabilizes itself in the complex environment is unclear. On the other hand, the activity of TpeL CPD was reported to be low[60,61] and TpeL is known to be least toxic in mice among LCTs[27].

It has been reported that the CROPs-truncated TcdA and TcdB underwent moderately increased InsP6-induced autocleavage compared to the full-length toxins[31,62]. The deletion of amino acids 1769-1787 caused spontaneous autocleavage of subtype 2 TcdB in the

absence of InsP6[63]. Here we demonstrate that the integrated CROPs domain forcibly protects TcsL from autoproteolysis by at least two mechanisms: the N-terminus of TcsL CROPs strongly restrains the autoproteolysis while longer CROPs further reduce the InsP6-induced autocleavage.

The cryo-EM structure of TcsL at neutral pH is an open conformation and similar to the previously reported structures of TcdB variants[31–33]. Given the high sequential consensus between TcsL and TcdB, this result is somewhat expected. Intriguingly, cryo-EM structures of the full-length TcsL in both the open and closed conformations were captured in high resolution at pH 5.0.

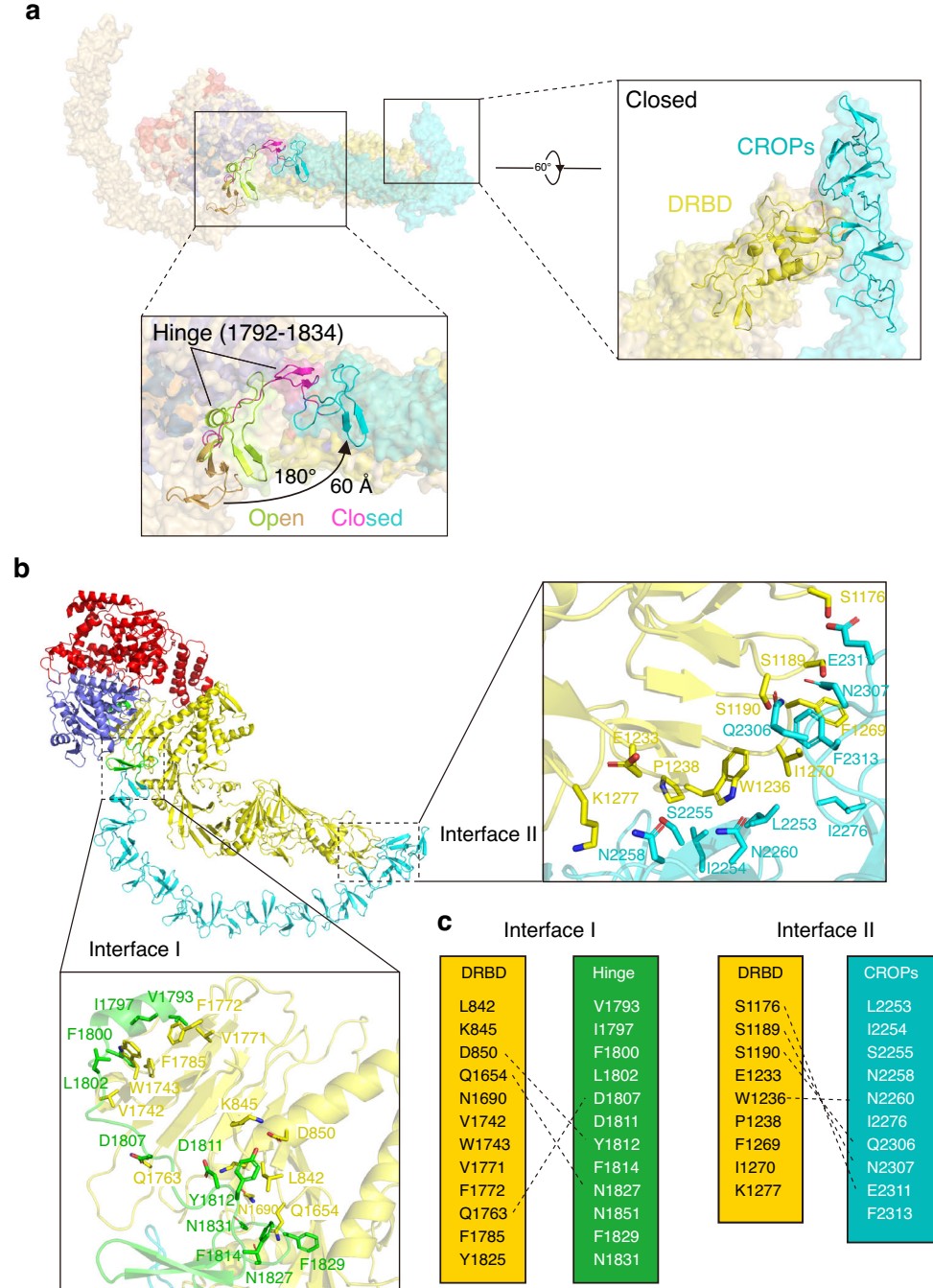

**Fig. 4 | Comparison of the TcsL structures between open and closed states.**
**a** Superposition of TcsL at pH 5.0 (closed state, colored as in Fig. 3.) and at pH 7.4 (open state, in light orange). Close-up view of the major structural changes of the CROPs domain between. The allosteric transition between the hinge region (residues 1792-1834) in the open state TcsL (lemon and brown) and the closed state TcsL (magenta and cyan) causes the rotation of the first SR of CROPs for -180°, -60 Å, leading to the CROPs unit-IV touches the "tip" part of the DRBD. **b** The close-up view of two intra-molecular interfaces. Interface I expanding across CPD-DRBD to CROPs unit-I. Interface II among the CROPs unit-IV and the middle part of DRBD (1150-1280) consists of several H-bonds and hydrophobic interactions. **c** Interfacing residues between the DRBD and CROPs. Dashed lines represent salt bridges.

Consistent with a previous report, the full-length TcsL is highly resistant to InsP6-induced proteolysis[54]. Two intramolecular interfaces between the CROPs and DRBD were observed in the closed-form structure of TcsL. The existence of the interface I may restrain the CPD and explain why the region located between residues 1808 to 1893 is important to prevent autocleavage of TcsL. The deletion of 19 residues (1848-1867) within this region would abolish the interactions of interface I. As expected, TcsL$^{\Delta 1848\text{-}1867}$ was simultaneously auto-cleaved during the protein expression. The emergence of interface II (between DRBD and CROPs unit IV) in the closed conformation may further lock the CPD and CROPs domain, as C-terminally truncated TcsL showed increased InsP6-induced autoproteolysis compared to the TcsL. Functional analyses on the chimeric toxin TcsL$^{2261B}$ provide us with stronger evidence to demonstrate the role of interface II. TcsL$^{2261B}$ contains a TcdB CROPs unit IV thus interactions in interface II are affected. As a result, TcsL$^{2261B}$ showed stronger InsP6-induced autocleavage compared to the WT toxin.

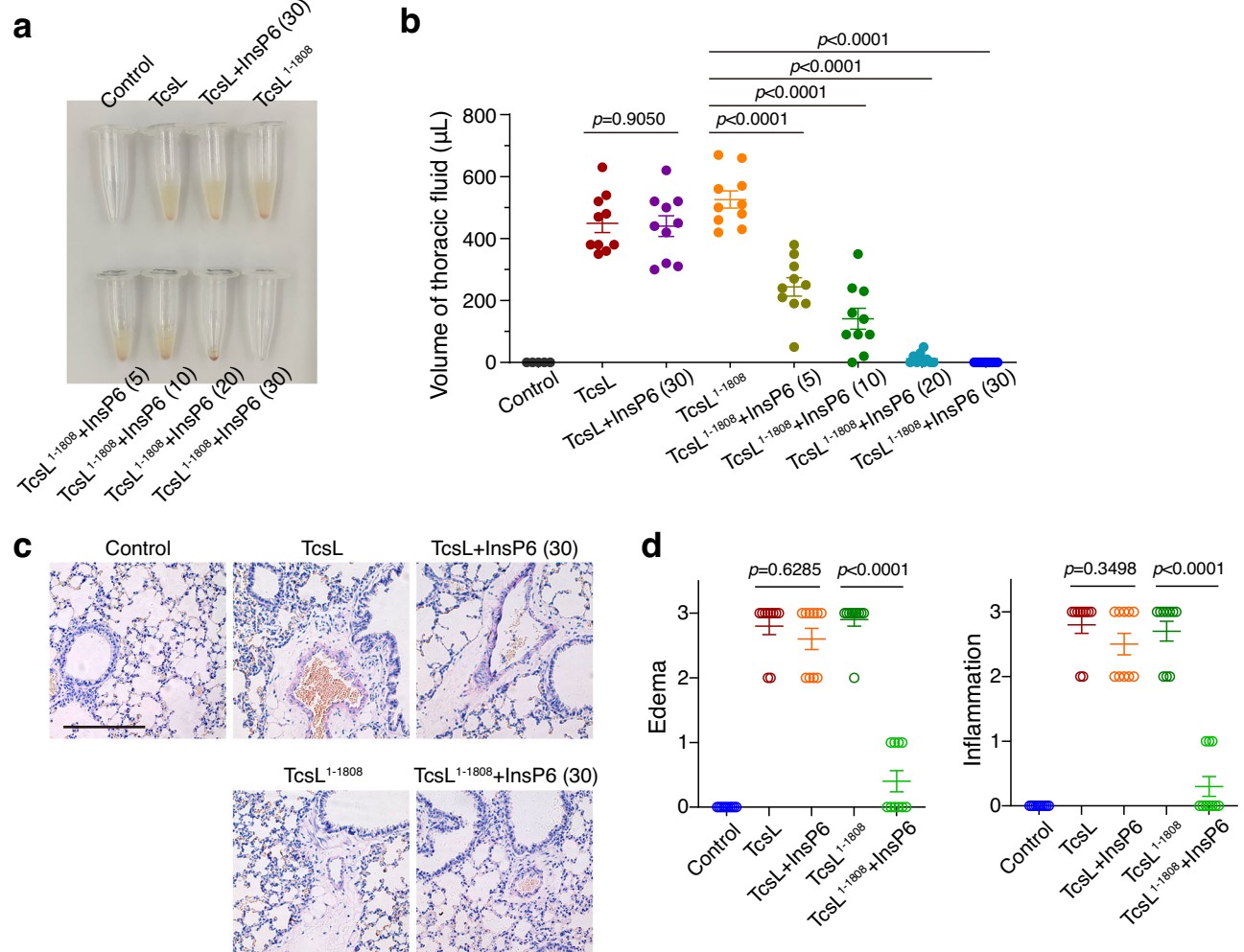

**Fig. 5 | CROPs retain toxicity of TcsL on mouse lung tissues. a** TcsL or TcsL[1–1808] was pre-incubated with InsP6 for the indicated time (in parentheses) and then IP-injected into the mice. After four hours, fluid was collected from the mice's thoracic cavity. The representative fluid accumulations are shown. **b** The volume of fluid collected from each mouse was measured and plotted. Error bars indicate mean ± SEM, *n* = 10 mice per group, two-tailed Mann-Whitney test. **c** H&E staining showing the lung tissue sections from mice injected with 0.8 μg/kg TcsL or 0.6 μg/kg TcsL[1–1808] pre-incubated with or without 100 μM InsP6. The scale bar represents 200 μm. **d** Histopathological scores for (**c**) were assessed based on indicated pathological features. Error bars indicate mean ± SEM, *n* = 10 mice per group, two-tailed Mann-Whitney test. Source data are provided as a Source Data file.

In the closed conformation, the CROPs domain of TcsL may sterically hinder the binding of the toxin receptor SEMA6A/6B (Supplementary Fig. 9). This could in part explain why TcsL has reduced cytotoxicity than C-terminal truncated TcsL in A549 cells. However, TcsL[1–2260] and TcsL[2261B] are also more potent than the full-length TcsL to HeLa cells, which express little to no SEMA6A/6B[48], implying TcsL might have additional uncharacterized receptor(s).

Based on these findings, we propose a CROPs-mediated self-stabilizing strategy for TcsL and likely other LCTs in complex extracellular environments. The closed conformation toxin could be tighter and more resistant to autoproteolysis and degradation, as additional interactions between the DRBD and CROPs were observed. The open conformation TcsL is more suitable to attack the host cells as its receptor binding region for SEMA6A/B is fully exposed. After endocytosed by the target cells, the CPD is translocated to the cytosol while the CROPs stay in the lumen of the acidic endosome. In this case, the CROPs and CPD are located on either side of the endosomal membrane; and InsP6-induced and CPD-mediated autoproteolysis would conduct normally (Fig. 7). Taken together, our study renovates the current understanding of the intrinsic functions of the CROPs domain and intramolecular regulatory mechanisms of TcsL. The structural and

functional insights into the CROPs-dependent autoproteolysis inhibition also expose vulnerabilities of TcsL (and possibly other LCTs), which can be utilized to develop potential therapeutic avenues against LCT-related diseases.

## Methods

### Ethics statement

All procedures were conducted following the guidelines approved by the Institutional Animal Care and Use Committee at Westlake University (IACUC Protocols #19-010-TL and #22-018-TL-5). To minimize the pain and distress, mice were monitored every hour after the toxin injection. Animals with signs such as labored breathing, inability to move after gentle stimulation, or disorientation were euthanized immediately. This method was approved by the IACUC and monitored by a qualified veterinarian.

### Materials

HeLa (H1, CRL-1958) and A549 (CCL-185) cells were originally obtained from ATCC. They were tested negative for mycoplasma contamination and authenticated via STR profiling (Shanghai Biowing Biotechnology Co. LTD, Shanghai, China). Cells were cultured in DMEM medium plus

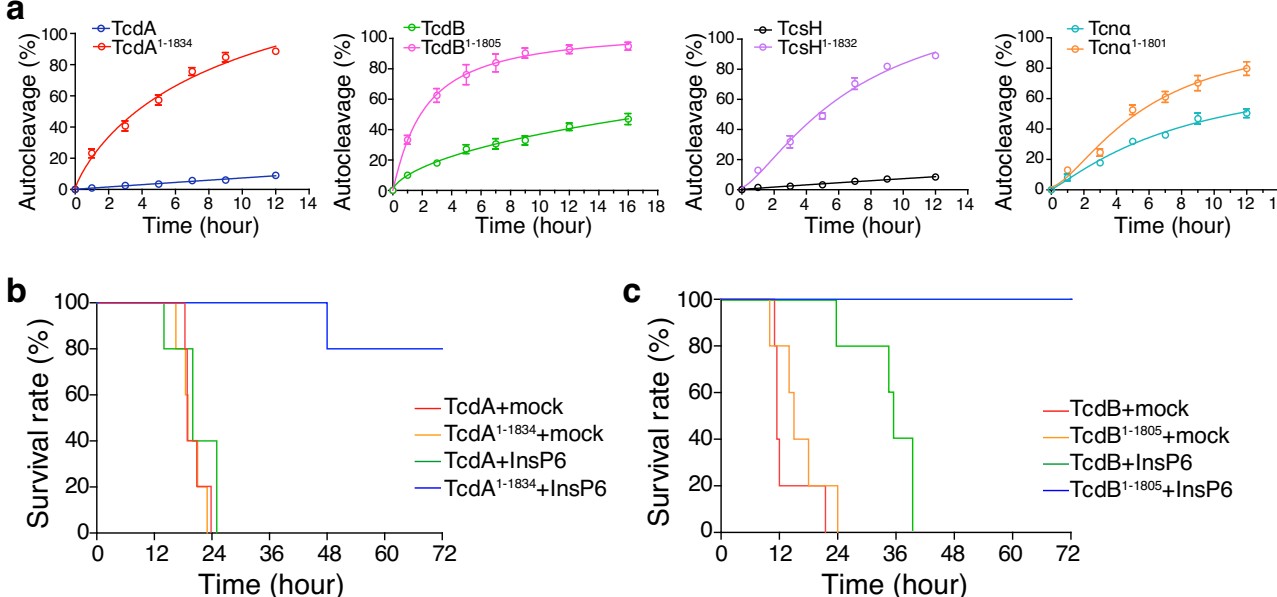

**Fig. 6 | CROPs-mediated inhibition on autoproteolysis of TcdA and TcdB. a** In the presence of 100 µM InsP6, the time course cleavage revealed increased autoproteolysis in the CROP-less LCTs (TcdA$^{1-1834}$, TcdB$^{1-1805}$, TcsH$^{1-1832}$, and Tcnα$^{1-1801}$) compared to the full-length toxins. Error bars indicate mean ± SD, $n = 3$ independent experiments. **b** 2 µg/kg TcdA and 1.35 µg/kg TcdA$^{1-1834}$ were pre-incubated with or without 100 µM InsP6 for 30 minutes and then IP injected into the mice. The survival of the mice is shown by the Kaplan-Meier curves, $n = 5$ mice per group. **c** 1 µg/kg TcdB and 0.75 µg/kg TcdB$^{1-1805}$ were pre-incubated with or without 100 µM InsP6 for 30 minutes and then IP injected into the mice. The survival of the mice is shown by the Kaplan-Meier curves, $n = 5$ mice per group. Source data are provided as a Source Data file.

10% fetal bovine serum (FBS) and 1% penicillin/streptomycin in a humidified atmosphere of 95% air and 5% CO2 at 37°C. InsP6 (#S3793) was purchased from Selleck.

### Mice
BALB/c mice (6-8 weeks, male, specific-pathogen-free) were purchased from the Laboratory Animal Resources Center at Westlake University (Hangzhou, China). Mice were housed with food and water without limitation and monitored under the care of full-time staff.

### Cloning, expression, and purification of recombinant proteins
DNA fragments encoding TcsL (reference sequence: *P. sordellii* 9048), TcdB1 (reference sequence: *C. difficile* 630), Tcnα (reference sequence: *C. novyi* GD211209) were codon-optimized, synthesized by Genscript (Nanjing, China). DNA fragments encoding the full-length and truncated toxins were PCR amplified and cloned into a modified pHT01 vector. Point mutations and deletion were performed using a Quick-change II Site-Directed Mutagenesis Kit (#200523, Agilent Technologies) following the manufacturer's instructions. C-terminal His-tagged recombinant proteins, including TcsL, TcsL$^{1-1808}$, TcsL$^{1-1893}$, TcsL$^{1-1921}$, TcsL$^{1-1977}$, TcsL$^{Δ1848-1867}$, TcsL$^{1-1808/C698S}$, TcsL$^{Δ1848-1867/C698S}$, TcsL$^{2261B}$, TcdB, TcdB$^{1-1805}$, Tcnα, and Tcnα$^{1-1801}$ were expressed and purified from *Bacillus subtilis* strain SL401. *B. subtilis* cells were cultured at 37 °C till OD$_{600}$ reached 0.6 and then induced with 1 mM isopropyl-β-D-thiogalactoside at 25 °C for 20 hours. All recombinant proteins were purified by Ni-affinity chromatography, followed by size-exclusion chromatography (GE Healthcare). The purified TcsL was further applied to the size-exclusion chromatography at a neutral pH using the buffer (20 mM tris-Cl pH 7.4, 150 mM NaCl) or at an acidic pH using the buffer (20 mM sodium acetate pH 5.0, 150 mM NaCl) and the peak fractions were applied to the cryo-EM sample preparation.

### Cryo-sample preparation and EM data collection
Each four µL aliquots of the purified TcsL at pH 7.4 and pH 5.0 were applied onto Holey carbon grids (Quantifoil, Au, 300-mesh, R1.2/1.3), which were glow-discharged in the Plasma Cleaner (HARRICK PLASMA Company). The grids were then blotted for 3.5 seconds and quickly plunged into liquid ethane cooled by liquid nitrogen using Vitrobot Mark IV (Thermo Fisher) at 8 °C and 100% humidity.

The cryo-grids were transferred to a 300-kV Titan Krios electron microscope (Thermo Fisher Scientific) equipped with a Gatan K3 detector and GIF Quantum energy filter (slit width 20 eV). The micrographs with the preset defocus range from −1.5 to −2.0 µm were collected at a normal magnification of 81,000× in super-resolution mode. Each stack with 32 frames was exposed for 2.56 s with a total dose of ~50 e-/Å² using EPU (Thermo Fisher Scientific). The movies are aligned and summed using MotionCor2 with a binning factor of 2, resulting in a pixel size of 1.087 Å[64]. Dose weighting was performed concurrently and the defocus value for each image was determined by Gctf[65] (Supplementary Table S1).

### Cryo-EM data processing
The cryo-EM data processing procedures for the TcsL at pH 7.4 were mainly carried out in RELION 3.0[66] except that is specially mentioned. First, 1,217,203 particles were auto-picked from 1,862 images using Gautomatch (developed by Kai Zhang, https://www2.mrc-lmb.cam.ac.uk/download/gautomatch-053/). We then performed the 2D classifications on the total particles and used part of the good particles to generate the initial models. After 2D selection, 637,210 particles were further applied to 3D classifications. Finally, we generated the reconstruction of TcsL at pH 7.4 at an average resolution of 2.9 Å.

The cryo-EM data processing procedures for the TcsL at pH 5.0 were mainly carried out in cryoSPARC[67]. A total of 12,265 micrographs were collected using the same condition as that of TcsL at neutral pH. About 12 million particles were generated from 11,464 manually selected micrographs. After 2D classifications, about 3.3 million particles were selected and two distinct features were observed. Following the initial model generation, the Heterogeneous refinement was applied to the remaining particles. The two conformations of the

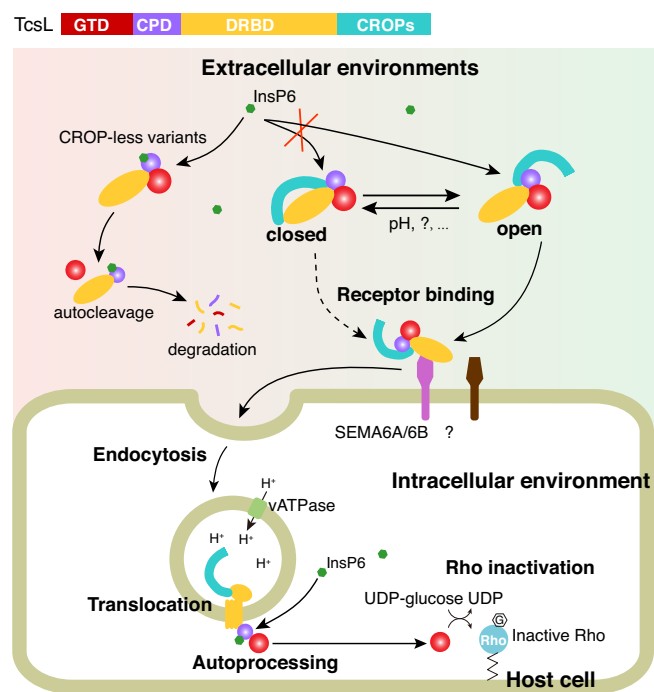

**Fig. 7 | Proposed model for CROPs-mediated self-stabilization of TcsL.** In the complex extracellular environments, TcsL may exist in either open or closed conformation. TcsL in the open conformation better attacks the host cells, while the closed-form TcsL might be more resistant to autoproteolysis and degradation. The CROP-less toxin variants are sensitive to InsP6 and will easily undergo autoproteolysis and degradation. Once enter the host cell via endocytosis, the toxins undergo conformational changes in the acidic endosomes and translocate the GTD and CPD into the cytosol. Thus, the CROPs would no longer inhibit the InsP6-induced autocleavage.

closed and open TcsL at an acidic pH were finally reconstructed at an average resolution of 2.9 Å and 2.5 Å using 410,049 and 1,045,888 particles, respectively.

The reported resolutions are calculated based on the FSC value of 0.143[68]. The angular distributions of the particles used for the final reconstructions are reasonable. Local resolution variations for the TcsL at pH 7.4 and pH 5.0 were estimated using Resmap[69] and cryoS-PARC, respectively. The EM density maps display clear features that allow detailed structural examination of protein sidechains (Supplementary Table S1).

### Model building and refinement
The atomic models were mainly generated by de novo modeling facilitated by AlphaFold[70]. The predicted structure was fitted into the individual EM density map for the TcsL at pH 7.4 or 5.0 using Chimera[71] and manually adjusted using Coot[72]. The final models of the TcsL at pH 7.4 and pH 5.0 were refined using the phenix.real_space_refine program in PHENIX with secondary structure restraints[73]. The structures were further validated through examination of the Molprobity scores and statistics of the Ramachandran plots. Molprobity scores were calculated as described[74] (Supplementary Table S1).

### Cytopathic cell rounding assay
Cells were seeded at a density of $3\times10^5$ cells per well in 24-well plates (Corning) and cultivated at 37 °C and 5% CO2 overnight. Toxins were then added into the culture medium in a serial dilution of 1/3 and incubated at 37 °C for 24 hours. The phase-contrast images of cells were then captured (Olympus IX73, 10× objectives). A zone of 200 × 200 μm was selected randomly, each containing ~20-50 cells. Round-shaped and normal-shaped cells were counted manually. All experiments were

performed in three independent biological replicates. Statistical analysis was performed using OriginPro v8.5 (OriginLab).

### Toxin challenge assay in mice
0.8 μg/kg TcsL, 0.6 μg/kg TcsL[1–1808], 2 μg/kg TcdA, 1.35 μg/kg TcdA[1–1834], 1 μg/kg TcdB, or 0.75 μg/kg TcdB[1–1805] was pre-incubated with or without 100 μM InsP6 in 200–500 μL of saline (0.9 % NaCl, pH 7.4) for 5, 10, 20, or 30 minutes. The mixtures were then IP-injected into the mice. For TcsL, all mice were euthanized after four hours. The fluid present in the thoracic cavity was collected for volume measurement and the lung segments of these mice were dissected for histopathological analysis. For TcdA and TcdB, the survival of the mice was illustrated by the Kaplan-Meier curves (monitored for 72 hours).

### H&E staining and histopathological analysis
Lung specimens were fixed in formalin for 12 hours before being dehydrated with an alcohol gradient, cleared with xylene, and then embedded in paraffin. Paraffin blocks were cut into 5 mm thick sections. The sections were stained with H&E. The H&E staining sections were scored blinded by two pathologists based on edema and inflammatory cell infiltration on a scale of 0 to 3 (mild to severe). The average scores were plotted on the charts.

### Statistics and reproducibility
Data are presented as mean ± standard deviation (SD) for biochemical experiments and mean ± standard error of the mean (SEM) for pathological experiments. The number of the sample size (n) and statistical hypothesis testing method are described in the legends of the corresponding figures. Statistical analyses of data were performed with GraphPad Prism v9.3 or OriginPro v8.5. All experiments in Figs. 1b, 1d-e, and 1g have been repeated for at least three times with similar results.

### Reporting summary
Further information on research design is available in the Nature Portfolio Reporting Summary linked to this article.

## Data availability
The atomic coordinates for the TcsL at pH 7.4 and at the pH 5.0 in its open and closed conformations have been deposited in the Protein Data Bank (PDB) with the accession codes 8JB5, 8X2H, and 8X2I, respectively. The EM maps have been deposited in the EMDB with the accession codes EMD-36141, EMD-38010, and EMD-38011 for the TcsL at pH 7.4 and 5.0 in its open and closed conformations, respectively. The other data that support the findings of this study are available in the publicly accessible repository. Source data are provided with the paper. Source data are provided with this paper.

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

## Acknowledgements

We thank Drs. Min Dong and Peng Chen for the suggestion and discussion. We also appreciate the technical support from the Biomedical Research Core Facility, High-Performance Computing Center, and Laboratory Animal Resources Center at Westlake University. This study was partially supported by the National Key R&D Program of China (Grant no. 2023YFC2308400 to L.T.), the National Natural Science Foundation of China (Grant no. 31970129 to L.T.), the Zhejiang Provincial Natural Science Foundation of China (Grant no. LR20C010001 to L.T. and Grant no. LQ23C010001 to J.L.), Postdoctoral Innovation Talents Support Program (BX20230325 to Y.Z.), and the Westlake Center for Genome Editing (Program no. 21200000A992210 to L.T.). L.T. also acknowledges support from the Westlake Laboratory of Life Sciences and Biomedicine and the Westlake Education Foundation.

## Author contributions

L.T. conceived the project and designed the experiments. Y.Zhou, D.L., Z.P., and Y.Zhang performed microbiological, biochemical, and cell biological assays. X.Zhan, R.Z., J.Z., X.Zhang, and Y.L. performed protein purification, cryo-EM structure determination, and structural data analysis. Y.Zhou, J.L., and T.J. performed animal experiments and pathological analyses. X.Zhan and L.T. wrote the manuscript with input from all co-authors.

## Competing interests

The authors declare no competing interests.
