## [Peer Review File · Nature Communications]

REVIEWER COMMENTS

Reviewer #1 (Remarks to the Author):

The lethal toxin (TcsL) produced by *Paenoclostridium sordellii* can cause lethal toxic shock syndrome and poses a potential life-threatening risk. It belongs to the large clostridial toxin (LCT) family and they all share a similar 4-domain architecture. In this study conducted by Zhou et al., it was observed that the CROPs domain of TcsL plays a crucial role to modulate its sensitivity to InsP6-dependent autoproteolysis and cytotoxicity. The authors also reported the cryoEM structures of TcsL in the presence and absence of InsP6 at a neutral pH. They found that the apo TcsL adopts an open conformation, resembling the structure of TcdB at acidic pH, and TcsL in the presence of InsP6 adopts a closed conformation, akin to the structure of TcdA at neutral pH. Largely based on these findings, the authors suggest that InsP6 induces the conformational change of TcsL to the closed state, which “locks the cysteine protease domain and likely hinders the binding of the host receptor.”

However, these results contradicted with a 2011 paper that reported a comprehensive study on InsP6-dependent processing of TcsL (PMID: 21385871). In the 2011 paper, Aktories and colleagues demonstrated that autocatalytic cleavage and InsP6 binding of full-length TcsL depended on acidic pH, and they observed no autocatalytic cleavage for the full length TcsL with InsP6 at neutral pH in vitro. Oddly, this paper was not cited/discussed in this manuscript. Furthermore, extensive research has been conducted on the impact of CROPs on cytotoxicity of LCTs, and it is widely believed that CROPs domain plays a crucial role in inhibiting autoproteolysis and preventing premature autoproteolysis. These prior arts further diminish the novelty of this research. Therefore, this manuscript does not represent the type of advance suitable for Nature Communications. Some more specific comments are listed below.

Major issues

1. The InsP6-dependent self-stabilizing strategy of TcsL proposed here is very puzzling. If TcsL needs InsP6 to take the closed conformation at neutral pH in order to inhibit the CPD activity and sterically hinder the binding of its host receptor, when, where and how will TcsL switch to the open conformation to attack host cells, considering that InsP6 is always around and probably even more inside the cells? In contrast, Aktories suggested that an acidic pH is needed to trigger the conformational change so TcsL will be able to bind InsP6. Therefore, it is crucial for the authors to carefully examine and compare the behavior and structure of TcsL at both acidic and neutral pH.
2. The chimeric protein TcsL2261B is interesting, which could help verify some of the findings reported here. For instance, does TcsL2261B take an open or closed conformation with or without InsP6 at

neutral (or acidic pH)? How is the InsP6-triggered cleavage pattern different between TcsL2261B and the WT toxin?

3. For the cryoEM study, what was the final pH of the apo and InsP6-incubated TcsL solution? What's the % of open and closed conformations observed with or without InsP6 among all particles picked?

4. The biology and the structure parts of this manuscript do not seem to help each other. For example, do the 3D structures reveal any hints about how InsP6 triggers the closed conformation? This is arguably the most important question in this story.

Reviewer #2 (Remarks to the Author):

Paenicostridium sordellii lethal toxin (TcsL) is a potent exotoxin that causes the lethal toxic syndrome. The author reported that TcsL with varied lengths of combined repetitive oligopeptides (CROPs) deleted showed increased autoproteolysis and higher cytotoxicity. Next, the author presents cryo-EM structures of full-length TcsL in apo and inositol hexakisphosphate (InsP6) at neutral pH. The apo exhibits open conformation, which resembles reported TcdB structures. On the other hand, InsP6 structure would induce the conformational change to the closed conformation.

Major points:

The author got the two significant conformations open and closed without and with InsP6.

Closed conformation of TcdA and open conformation of TcdB were observed. However, both conformations could be seen in TcsL, and it is exciting that it depends on InsP6.

However, the next points are not known from this paper.

(1) How InsP6 can regulate the two conformations.

(2) How closed conformation upregulates autoproteolysis using the CPD domain.

(3) What is the conformation in the extracellular or endocytosis environment?

(4) Open conformation without InsP6 is stable: I can not understand this.

Open conformation seems unstable.

Furthermore, all processing steps were carried out by cryoSPARC. The author classified two conformations. CryoSPARC can trap the successive density from close to open state. Please add this processing results.

Minor points

autoprocessing>

Is the autoprocessing occur by the same molecule or the other molecule?

CROPs-mediated autoprocessing>

InsP6 will determine the CROP's orientation. Ins-P6 induces closed conformation.

Without Ins-P6, it is open conformation. It seems to be good for autocleavage by CPD.

If so,

Fig1g High Ins-P6 concentration will increase autoprocessing?

I do not understand this.

Autoprocessing should be high in the open conformation.

Fig1h CROP less mutant will increase autoprocessing activity.

I am not sure why CROP less mutant is high. CROP less mutant is accessible by other molecule's CPD. Please explain this.

Fig4b

The beta-flap structure of CPD in the full-length TcsL remains unchanged upon Ins6P induction: So, furthermore, I do not understand the reason why closed conformation is better for autoprocessing.

L108

TcsL CROPS can specifically suppress the cytotoxicity of TcsL.

> It is interesting, but please explain the reason.

Fig3b

InsP6 induced closed conformation: It is too important to show the electron density of InsP6.

L206 CPD-bound InsP6> Why does the author use this expression? I am not sure. Is there any fact to bind to CPD?

L235

CROPs-dependent autoproteolysis inhibition could be functionally significant for maintaining the toxicity of TcsL before host invasion.

Please add the summary figure to express the toxicity of TcsL.

Maybe, it is better to add an explanation.

open> close(InsP6)> autoprocessing> toxicity

Is this related to the pore-forming across the endosome membrane?

If so, add the figure.

L246

LCTs need to prevent environmental factors (InsP6) -induced autoproteolysis and degradation.> Again, I do not understand the mechanism to inhibit proteolysis without CROPs.

L247

a brief five-minute pre-incubation of TcsL1-1808 with InsP6 significantly reduced the majority of toxin activity.

> I also do not understand how to increase proteolysis by insP6-induced CROPs conformational change.

L269

CPD remains in relatively inactive state even upon InsP6 induction>

Compared with the other Cysteine protease, is there any structural feature in an inactive state?

L286

In complex extracellular environments, TcsL mainly exists in the closed conformation >

Is this right? Is this open conformation without InsP6 in extracellular environments?

Again, open conformation is stable for protease, isn't it? Please use the summary figure and add a simple explanation.

Reviewer #3 (Remarks to the Author):

General Overview and Comments:

The role of the CROPs domain in the family of Large Clostridial Toxins (LCTs) has in recent years come into question. Previously thought to function solely in host recognition, the CROPs domain has now been shown to be dispensable in host receptor binding, suggesting this domain may have alternate functions in LCT activity. Thus, the authors sought to investigate the role of the CROPs domain in the *Paenibacillus sordellii* LCT, TcsL, which shows high sequence identity to *Clostridioides difficile* TcdB.

Key Results

- Purification of full length and truncated versions of TcsL identified that the CROPs domain plays an important role in protecting the full-length toxin from proteolytic activity/auto-proteolysis. Through directed and specific truncation of TcsL it was determined that residues 1848 to 1867 tightly regulate

the auto-cleavage of TcsL – which may have implications in understanding the cleavage of other LCT members. This was partially supported with data from full length and truncated *C. difficile* TcdB and *Clostridium novyi* Tcn α , which despite being stable as their truncated forms (in contrast to TcsL), were still more sensitive to auto-cleavage.

- Truncation of TcsL lead to increase cytotoxicity in vitro, suggesting that CROPs in part dictates the virulence capacity of TcsL. This contrasts previous work that suggests truncation of *C. difficile* TcdA, TcdB, and *P. sordellii* TcsH reduces virulence capacity.

- To investigate the mechanism behind these changes in auto-cleavage and virulence capacity, CryoEM was employed to understand the confirmation of full length TcsL. These studies showed that TcsL adopts an “open” confirmation, that switches to a closed confirmation in the presence of Inositol hexakisphosphate (InsP6), which induces auto-cleavage of LCTs. Thus, the CROPs may aid in stabilising and protecting TcsL from auto-cleavage. To test the functional consequences of this effect in vivo, mice were exposed to TcsL or a truncated TcsL with and without pre-exposure to InsP6. Under normal condition, TcsL leads to thoracic fluid accumulation and lung damage, a response that was unaltered by exposure to InsP6. In contrast, while truncated TcsL alone could induce lung damage, this was abrogated by exposure to InsP6, suggesting that the CROPs aid in maintaining virulence capacity in vivo.

Conclusions

- Together, these data highlight an important role for CROPs in protecting TcsL from auto-cleavage and in maintaining virulence capacity.

- As TcsL has high similarity to other LCTs these functions may be conversed across the LCTs.

Concerns/Originality

- Convincing data to support the theory that CROPs may have a protective role across the LCTs is not provided or entirely warranted. While the findings within this work are interesting, it is unclear if there is a significant shift in knowledge for the field. While *P. sordellii* infections are highly fatal, the authors note that they are quite rare. Therefore, the impact of this work may not be far reaching, unless a similar phenotype is seen across of LCTs, such as the *C. difficile* toxins, TcdA and TcdB, which have a broader impact on human and animal disease. The authors purpose that the CROPs leads to a self-stabilising of TcsL, which they believe is likely for other LCTs. However, data from the literature is presented that suggests the other LCTs do not all function in similar ways and have shown inconsistent results with regards to the CROPs domain. Additionally, TpeL from *C. perfringens* lacks a CROPs domain, but still produces functional toxin.

Suggestions

· Given data was presented in this manuscript for full length and truncated *C. difficile* TcdB and *Clostridium novyi* Tcn α (with TpeL unsuitable for this work as it lacks a CROPs domain), it may be beneficial for the authors to investigate the role of CROPs in these toxins (and if feasible TcdA and TcsH), both in vitro and in vivo to further support this claim (several intestinal injection models have been published for *C. difficile* toxins for example, and cell based assays exists for TcdA, TcdB, TcsH). However, as it stands, I do not believe that the work presented here represents an important advancement of knowledge that is of significance to the Clostridial field.

Response to Reviewers (NCOMMS-23-26722-T)

We appreciate the reviewers and editors for their constructive suggestions and comments. Following their suggestions, we have revised our manuscript and made the point-to-point response as below.

Reviewer #1

The lethal toxin (TcsL) produced by Paeniclostridium sordellii can cause lethal toxic shock syndrome and poses a potential life-threatening risk..... However, these results contradicted with a 2011 paper that reported a comprehensive study on InsP6-dependent processing of TcsL (PMID: 21385871). In the 2011 paper, Aktories and colleagues demonstrated that autocatalytic cleavage and InsP6 binding of full-length TcsL depended on acidic pH, and they observed no autocatalytic cleavage for the full length TcsL with InsP6 at neutral pH in vitro. Oddly, this paper was not cited/discussed in this manuscript.....

Response: Thank you for the comment. We are sorry for not citing this closely related literature. We have included this reference in our manuscript. On the other hand, we think that their results are not essentially contradictory to ours. Indeed, both studies suggest that the full-length TcsL is highly resistant to InsP6-induced autoproteolysis. We have also updated Figure 1g and Supplementary Fig. 1. (Line 139-141, 315-317, Ref. #54)

Major issues

1. The InsP6-dependent self-stabilizing strategy of TcsL proposed here is very puzzling. If TcsL needs InsP6 to take the closed conformation at neutral pH in order to inhibit the CPD activity and sterically hinder the binding of its host receptor, when, where and how will TcsL switch to the open conformation to attack host cells, considering that InsP6 is always around and probably even more inside the cells? In contrast, Aktories suggested that an acidic pH is needed to trigger the conformational change so TcsL will be able to bind InsP6. Therefore, it is crucial for the authors to carefully examine and compare the behavior and structure of TcsL at both acidic and neutral pH.

Response: Thank you for the comments and suggestions. The extracellular environments vary. In the presence of high levels of InsP6 (such as sometimes in the soil and feces), TcsL exists in the close conformation to prevent autoproteolysis and thus stays intact. If there's little to no InsP6 (for example, in the soft tissues or blood), TcsL would be in the open conformation and attack the host cells. When the toxin enters the host cell, the situation is different. TcsL undergoes conformational changes in the acidic endosome to translocate its GTD and CPD into the cytosol. We would like to emphasize two points: (1) This transmembrane conformation/structure of TcsL (and for all LCTs) is still unknown; it is generally considered that the CROPs and CPD are

located at either side of the endosomal membrane. (2) The InsP6-induced autocleavage occurs in the cytosol where the pH is neutral. Thus, InsP6-induced autocleavage under acidic pH is less physiologically relevant. To better express our views, we have added an illustrative scheme to our manuscript. (Line 341-351, Figure 8)

We also agree with the reviewer that examining the structure of TcsL at acidic pH could be interesting. Following the suggestion, we have resolved the cryo-EM structure of TcsL at pH=5.0. The structural analysis demonstrates that TcsL simultaneously exhibits two distinct classes at acidic pH with both open (~53.8%) and closed (~46.2%) conformations. Low pH would induce a conformational switch of TcsL from open to closed state, which is also partly supported by a previous XL-MS analysis for TcdB. (Line 226-234, Figure 5d, Supplementary Fig. 7)

2. The chimeric protein TcsL2261B is interesting, which could help verify some of the findings reported here. For instance, does TcsL2261B take an open or closed conformation with or without InsP6 at neutral (or acidic pH)? How is the InsP6-triggered cleavage pattern different between TcsL2261B and the WT toxin?

Response: We appreciate the reviewer for these comments. We have performed the single-particle cryo-EM analysis on the chimeric protein TcsL^{2261B} at the neutral pH. Through 2D analysis, all TcsL^{2261B} protein particles identified took an open conformation both in the absence or presence of InsP6 at the pH of 7.4, which is different from the WT toxin. Consistent with the structural finding, TcsL^{2261B} exhibits a stronger InsP6-induced autocleavage compared to the WT toxin. (Line 170-174, 219-224, 327-333, Figures 2e and 5c, Supplementary Fig. 3)

3. For the cryoEM study, what was the final pH of the apo and InsP6-incubated TcsL solution? What's the % of open and closed conformations observed with or without InsP6 among all particles picked?

Response: In the initially submitted manuscript, both the apo and InsP6-incubated TcsL solutions used for cryo-EM studies were maintained at a neutral pH of 7.4. At this neutral pH, our 2D analysis revealed that we predominantly observed only open or closed conformation for the apo or InsP6-incubated TcsL. In the revised manuscript, we have expanded our investigation to include the structural analysis of apo TcsL at an acidic pH of 5.0. This adjustment in pH conditions led to a shift in the conformational distribution. Among picked particles for the final reconstructions, approximately 53.8% and 46.2% were observed in the open and closed conformations under the acidic pH, respectively. (Line 226-234, Figure 5d, Supplementary Fig. 7)

4. The biology and the structure parts of this manuscript do not seem to help each other. For example, do the 3D structures reveal any hints about how InsP6 triggers the closed conformation? This is arguably the most important question in this story.

Response: We appreciate your feedback and recognize the importance of establishing a strong connection between the biological and structural aspects of our manuscript.

We have repeatedly confirmed that no electron density for InsP6 was observed in our full-length TcsL structure in the presence of 100 μM InsP6 at pH7.4, demonstrating that InsP6 could hardly bind the CPD to induce autoproteolysis under this condition. (Line 312-318)

Reviewer #2

Major points:

The author got the two significant conformations open and closed without and with InsP6. Closed conformation of TcdA and open conformation of TcdB were observed. However, both conformations could be seen in TcsL, and it is exciting that it depends on InsP6.

Response: We very much thank the reviewer for the encouragement and support.

However, the next points are not known from this paper.

(1) How InsP6 can regulate the two conformations.

(2) How closed conformation upregulates autoproteolysis using the CPD domain.

(3) What is the conformation in the extracellular or endocytosis environment?

(4) Open conformation without InsP6 is stable: I can not understand this. Open conformation seems unstable.

Response: We appreciate the reviewer for the insightful comments. These points are all very interesting and related to each other.

The extracellular environment varies. If there's little or no InsP6, TcsL would be in the open conformation to attack host cells. If there's a high level of InsP6, TcsL would switch to the closed conformation to prevent the autoproteolysis and thus stay intact. We must confess that we don't know exactly how InsP6 triggers a conformational change from open to closed state. It was shown that InsP6 binds to the CPD alone with resolved structures (Pruitt et al., 2009, Shen et al. 2011). On the other hand, we have repeatedly confirmed that no electron density for InsP6 was observed in our full-length TcsL structure in the presence of 100 μ M InsP6 at pH7.4, demonstrating that InsP6 could hardly bind the CPD to induce autoproteolysis under this condition.

When TcsL enters the cell via endocytosis, the toxin undergoes a conformational change in the acidic endosome to translocate its GTD and CPD into the cytosol. This transmembrane conformation/structure of TcsL (and for all LCTs) is currently unknown and is a long-lasting question for the field. It is believed that the CROPs and GTD-CPD are located at either side of the endosomal membrane under this conformation. To better display these processes, we have added an illustrative scheme to our manuscript. (Line 312-318, 341-351, Figure 8)

Furthermore, all processing steps were carried out by cryoSPARC. The author classified two conformations. CryoSPARC can trap the successive density from close to open state. Please add this processing results.

Response: The open and closed conformations were obtained from samples in different conditions at the neutral pH. Notably, we could only capture one conformation in each cryo-sample. Thus, we could not trap the successive density using these samples.

In the revised manuscript, we have newly solved the structures of TcsL at an acidic pH of 5.0 and captured both the open and closed conformations. We tried to apply the good

particles from this data set to cryoSPARC, but we were unable to trap any successive density from closed to open state. Possibly, the middle states are just too transient to be captured. (Line 226-234, Figure 5d, Supplementary Fig. 7)

Minor points

autoprocessing> *Is the autoprocessing occur by the same molecule or the other molecule?*

Response: Based on previous literature, this is an intramolecular process. However, it seems that no direct evidence showing whether the autoprocessing would occur by other molecules.

CROPs-mediated autoprocessing>

InsP6 will determine the CROP's orientation. InsP6 induces closed conformation. Without InsP6, it is open conformation. It seems to be good for autocleavage by CPD. If so, Fig1g High Ins-P6 concentration will increase autoprocessing? I do not understand this. Autoprocessing should be high in the open conformation.

Response: We guess the reviewer's question is about the "CROPs-mediated autoprocessing regulation". Higher InsP6 concentration indeed increases the CPD-mediated autoprocessing. Also, InsP6 induces closed conformation, and the closed conformation inhibits the autoprocessing of TcsL. The key point is that all these happen in the extracellular environment. In the intracellular environment (the acidic endosomes), TcsL would turn into a transmembrane conformation that is currently unknown. It is considered that the CPD and CROPs are located on either side of the endosomal membrane. In this case, the CROPs-mediated autoprocessing regulation no longer exists. To better display the views, an illustrative scheme has been added to the manuscript. (Line 341-351, Figure 8)

Fig1h

CROP less mutant will increase autoprocessing activity. I am not sure why CROP less mutant is high. CROP less mutant is accessible by other molecule's CPD. Please explain this.

Response: We suggest that the CPD would be more accessible for InsP6 to bind and trigger proteolysis without the restraint of the CROPs domain. (Line 312-318)

Fig4b

The beta-flap structure of CPD in the full-length TcsL remains unchanged upon Ins6P induction: So, furthermore, I do not understand the reason why closed conformation is better for autoprocessing.

Response: The open conformation is better for autoprocessing, while the closed conformation is resistant to InsP6-induced autoprocessing. Sorry for the confusion, we have added a scheme to better explain it. (Figure 8)

L108

TcsL CROPS can specifically suppress the cytotoxicity of TcsL. > It is interesting, but please explain the reason.

Response: A possible reason is that the CROPS hinder receptor recognition. We have discussed it in the Discussion section. (Line 334-337)

Fig3b; InsP6 induced closed conformation: It is too important to show the electron density of InsP6. L206 CPD-bound InsP6> Why does the author use this expression? I am not sure. Is there any fact to bind to CPD?

Response: Thank you for the important comment. It was shown that InsP6 binds to the CPD alone with resolved structures (Pruitt et al., 2009, Shen et al. 2011). However, we have repeatedly confirmed that no electron density for InsP6 was observed in our full-length TcsL structure in the presence of 100 μ M InsP6 at pH7.4, demonstrating that InsP6 could hardly bind the CPD to induce autoproteolysis under this condition. (Line 312-318)

L235; CROPS-dependent autoproteolysis inhibition could be functionally significant for maintaining the toxicity of TcsL before host invasion. Please add the summary figure to express the toxicity of TcsL. Maybe, it is better to add an explanation. open> close(InsP6)> autoprocesing> toxicity. Is this related to the pore-forming across the endosome membrane? If so, add the figure.

L246; LCTs need to prevent environmental factors (InsP6) -induced autoproteolysis and degradation.> Again, I do not understand the mechanism to inhibit proteolysis without CROPS.

L247; a brief five-minute pre-incubation of TcsL1-1808 with InsP6 significantly reduced the majority of toxin activity. > I also do not understand how to increase proteolysis by insP6-induced CROPS conformational change.

Response: Thank you very much for the suggestions. Yes, the CROPS-mediated autoprocesing regulation happens in the extracellular environment. While in the acidic endosome, TcsL would present a transmembrane conformation that currently with no structure. In the transmembrane conformation, the CROPS and CPD are located at either side of the endosomal membrane, and thus no CROPS-mediated regulation on CPD exists. We strongly agree with the reviewer that it would be very important to include a summarized figure to explain these processes, which will largely help the readers to understand our views. As suggested, an illustrative scheme has been added to the manuscript. (Line 340-351, Figure 8)

L269

CPD remains in relatively inactive state even upon InsP6 induction> Compared with the other Cysteine protease, is there any structural feature in an inactive state?

Response: We have compared the CPD domain with other cysteine proteases but found no notable features. We have rephrased the wording for no ambiguity. (Line 308-309)

L286

In complex extracellular environments, TcsL mainly exists in the closed conformation >Is this right? Is this open conformation without InsP6 in extracellular environments? Again, open conformation is stable for protease, isn't it? Please use the summary figure and add a simple explanation.

Response: The extracellular environments vary. If there's no InsP6, TcsL would be in the open conformation. If there's a high level of InsP6, TcsL would switch to the closed conformation to prevent the autoproteolysis and thus stay intact. As suggested by the reviewer, an illustrative scheme has been added to the manuscript. (Line 340-351, Figure 8)

Reviewer #3

Suggestions

• *Given data was presented in this manuscript for full length and truncated C. difficile TcdB and Clostridium novyi Tcn α (with TpeL unsuitable for this work as it lacks a CROPs domain), it may be beneficial for the authors to investigate the role of CROPs in these toxins (and if feasible TcdA and TcsH), both in vitro and in vivo to further support this claim (several intestinal injection models have been published for C. difficile toxins for example, and cell based assays exists for TcdA, TcdB, TcsH). However, as it stands, I do not believe that the work presented here represents an important advancement of knowledge that is of significance to the Clostridial field.*

Response: We very much appreciate the reviewer for the comments and suggestions. We agree with the reviewer that showing more evidence for other LCTs, such as TcdA and TcdB, could be helpful. We have performed the InsP6-induced autocleavage assays *in vitro* for other full-length LCTs (TcdA, TcdB, TcsH, and Tcn α) and their CROP-less derivatives (TcdA¹⁻¹⁸³⁴, TcdB¹⁻¹⁸⁰⁵, TcsH¹⁻¹⁸³², and Tcn α ¹⁻¹⁸⁰¹). All tested LCTs in the full-length form showed weaker InsP6-induced autocleavage compared to their CROP-less versions, which is similar to the observation of TcsL and TcsL truncates. We also simulated the situation that TcdA and TcdB were pre-exposed to environmental InsP6 and then intoxicated the mice. When pretreated with extracellular InsP6, the CROP-less TcdA and TcdB, but not the full-length toxins, drastically lost their toxicity and became less potent in mice. Thus, the CROPs-mediated regulation on autoprocessing seems to be common among LCTs. (Line 261-277, Figure 7, Supplementary Fig. 8)

REVIEWER COMMENTS

Reviewer #1 (Remarks to the Author):

The authors did not meaningfully address my concerns, nor that from other reviewers. Frankly speaking, it's probably even more confusing now. For example, if "InsP6 could hardly bind the CPD", how "InsP6 would induce the conformational change of TcsL to the closed form"? Their new data also showed that TcsL could flip between the closed and open conformation irrelevant to InsP6. I suggest reject this manuscript so it can be considered for publication in another journal in a timely manner.

Reviewer #2 (Remarks to the Author):

The manuscript was mainly revised, including new cryo-EM experiment at acidic pH and the excellent Figure 8 to illustrate the scheme of the proposed model for CROPs-mediated regulation on autoproteolysis of TcsL. However, significant points were not addressed clearly.

Major points:

The author got the two significant conformations open and closed without and with InsP6. Closed conformation of TcdA and open conformation of TcdB were observed. However, both conformations could be seen in TcsL, and it is exciting that it depends on InsP6.

Response: We very much thank the reviewer for the encouragement and support. However, the next points are not known from this paper.

- (1) How InsP6 can regulate the two conformations.
- (2) How closed conformation upregulates autoproteolysis using the CPD domain.
- (3) What is the conformation in the extracellular or endocytosis environment?
- (4) Open conformation without InsP6 is stable: I can not understand this. Open conformation seems unstable.

Response: We appreciate the reviewer for the insightful comments. These points are all very interesting and related to each other.

The extracellular environment varies. If there's little or no InsP6, TcsL would be in the open conformation to attack host cells. If there's a high level of InsP6, TcsL would switch to the closed conformation to prevent the autoproteolysis and thus stay intact.

We must confess that we don't know exactly how InsP6 triggers a conformational change from open to closed state.

>

As the author concludes in the abstract that the presence of InsP6 would induce the conformational change of TcsL to the closed form, the InsP6 transformation from open to closed state is the key of the paper. Also, there is no indication of InsP6 density in the cryo-EM structure. Unfortunately, it is not sure whether the transformation from open to closed was induced by InsP6. Furthermore, the acidic pH cryo-EM structure exhibits two distinct conformations with both open (53.8%) and closed (~46.2%). Acidic pH did induce transformation.

It is still a big question how InsP6 can regulate the two conformations. Though there is unknown conformation in the endocytosis environment, I understand that InsP6 should be essential for the autocleavage. InsP6, not acidic pH, induces the closed conformation. Is this essential physiologically.

It was shown that InsP6 binds to the CPD alone with

resolved structures (Pruitt et al., 2009, Shen et al. 2011). On the other hand, we have repeatedly confirmed that no electron density for InsP6 was observed in our full-length TcsL structure in the presence of 100 μ M InsP6 at pH7.4, demonstrating that InsP6 could hardly bind the CPD to induce autoproteolysis under this condition.

>

Does InsP6 induce to the same closed conformation in other homologs?

CROPs-mediated autoprocessing>

InsP6 will determine the CROP's orientation. InsP6 induces closed conformation. Without InsP6, it is open conformation. It seems to be good for autocleavage by CPD. If so, Fig1g High Ins-P6 concentration will increase autoprocessing? I do not understand this. Autoprocessing should be high in the open conformation.

Response: We guess the reviewer's question is about the "CROPs-mediated autoprocessing regulation". Higher InsP6 concentration indeed increases the CPD-mediated

autoprocessing. Also, InsP6 induces closed conformation, and the closed

conformation inhibits the autoprocessing of TcsL. The key point is that all these happen in the extracellular environment.

In the intracellular environment (the acidic endosomes), TcsL would turn into a transmembrane conformation that is currently unknown. It is considered that the CPD and CROPs are located on either side of the

endosomal membrane. In this case, the CROPs-mediated autoprocessing regulation no longer exists. To better display the views, an illustrative scheme has been added to the manuscript. (Line 341-351, Figure 8)

>

New Fig.8 is very informative.

However, in the extracellular environment, InsP6 binding is the key to proving this hypothesis. Alternatively, the author may show the same InsP6-induced closed conformation in TcdA or TcdB without InsP6 binding. It is not enough, but is helpful to support their hypothesis.

The same InsP6 binding site is solvent accessible in the closed form compared with the CROP-less variant binding site?

Reviewer #3 (Remarks to the Author):

Peer Review: Structural dynamics of CROPs control the stability and toxicity of *Paenibacillus sordellii* lethal toxin

Liang Tao, Yao Zhou, Xiechao Zhan, Jianhua Luo, Diyin Li, Ruoyu Zhou, Jiahao Zhang, Zhenrui Pan, Yuanyuan Zhang, Tianhui Jia, Xiaofeng Zhang, and Yanyan Li (2023)

General Overview and Comments:

Following peer-review and manuscript edits, the authors have now presented new data and clarified concerns from each review regarding the role of CROPs in TcsL function and auto-processing.

Key New Results

- Previous work purified full length and truncated versions of TcsL and showed that the CROPs domain plays an important role in protecting the full-length toxin from proteolytic activity/auto-proteolysis when exposed to InsP6. In this revision, replacement of a region of TcsL with TcdB sequence led to increase resistance to auto-proteolysis when exposed to InsP6. These hybrid TcsL were always observed in the open confirmation via Cryo-EM
- Low pH resulted in a shift in the confirmation of TcsL, with a roughly 50:50 split in particles in the open or closed confirmation. This is important as within cells, TcsL is anticipated to undergo exposure to differing pH's which could alter its auto-proteolysis.
- Importantly, and as per suggestion, the protective effect of CROPs against auto-proteolysis was tested across the other LCTs. Each LCT was tested in full length and CROPs-less versions, with and without InsP6. As speculated, these LCTs showed similar protection against proteolysis when exposed to InsP6 when the full-length CROPs was intact.

- To test if this had a biological importance, TcdA and TcdB (full length and truncated CROPs-less versions) were tested in animals for toxicity through IP administration. Toxins were left untreated or treated with InsP6 and importantly showed that full-length toxins were able to induce disease, suggesting resistance to auto-proteolysis. In support of this idea, InsP6 treatment of CROPs-less toxins abrogated their toxicity.
- Through directed and specific truncation of TcsL it was determined that residues 1848 to 1867 tightly regulate the auto-cleavage of TcsL – which may have implications in understanding the cleavage of other LCT members. This was partially supported with data from full length and truncated *C. difficile* TcdB and *Clostridium novyi* Tcn α , which despite being stable as their truncated forms (in contrast to TcsL), were still more sensitive to auto-cleavage.

Conclusions

- Together, these data highlight an important role for CROPs in protecting TcsL, and the remaining LCTs (Particularly TcdA and TcdB) from auto-cleavage and in maintaining virulence capacity.

Concerns/Originality

- My previous concerns were regarding the assumption of CROPs playing a fixed role across the LCTs. I believe the authors have satisfied this concern, by purifying and testing TcdA, TcdB, TcsH, and Tcn α , and their truncated CROPs-less derivatives. Furthermore, issues regarding the application of this work in a human context for more prevalent disease (i.e. *C. difficile* infection, rather than *P. sordellii* infection which occurs significantly less frequently) has been addressed by the in vivo work assessing the toxicity of TcdA and TcdB and their truncated CROPs-less derivatives.
- My concerns regarding TpeL have not been addressed. I do understand that TpeL is a natural CROPs less LCT, but its use in this study would have proved valuable.

Suggestions

- With these revisions and data to support the protective phenotype of CROPs across the LCTs and in maintaining virulence capacity for TcdA and TcdB, as well as TcsL, the work presented here represents an important advancement of knowledge of significance to the Clostridial field.
- I am willing to accept these modifications for publication. However, the authors may want to comment on how this data relates to TpeL which lacks a functional CROPs domain.

Response to Reviewers (NCOMMS-23-26722-B)

We very much thank the reviewers and editors for their comments and suggestions. As suggested, we have revised the manuscript and made the response to reviewers as below.

Reviewer #1

Frankly speaking, it's probably even more confusing now. For example, if "InsP6 could hardly bind the CPD", how "InsP6 would induce the conformational change of TcsL to the closed form"? Their new data also showed that TcsL could flip between the closed and open conformation irrelevant to Insp6.

Response: We thank the reviewer for the comments. We agree that conclusions relating to whether and how InsP6 induces conformational changes in TcsL could be confusing and controversial. Following the editors' suggestions and requests, we have removed the related data and statements from the manuscript.

Reviewer #2

> As the author concludes in the abstract that the presence of InsP6 would induce the conformational change of TcsL to the closed form, the InsP6 transformation from open to closed state is the key of the paper. Also, there is no indication of InsP6 density in the cryo-EM structure. Unfortunately, it is not sure whether the transformation from open to closed was induced by InsP6. It is still a big question how InsP6 can regulate the two conformations. Though there is unknown conformation in the endocytosis environment, I understand that InsP6 should be essential for the autocleavage. InsP6, not acidic pH, induces the closed conformation. Is this essential physiologically.

> Does InsP6 induce to the same closed conformation in other homologs?

> New Fig.8 is very informative. However, in the extracellular environment, InsP6 binding is the key to proving this hypothesis. Alternatively, the author may show the same InsP6-induced closed conformation in TcdA or TcdB without InsP6 binding. It is not enough, but is helpful to support their hypothesis. The same InsP6 binding site is solvent accessible in the closed form compared with the CROP-less variant binding site?

Response: We appreciate the reviewer for the discussions, comments, and suggestions, which are insightful. We realize that conclusions relating to the ability of InsP6 regulation to induce conformational changes in TcsL could be controversial and confusing. Given this, the editor suggested and asked us to remove the related sections from our manuscript and we are happy to do so. Follow the editor's request, we have removed these data and made some other related changes as follows: (1) the previous Figs. 4a, 4b, and 5c were removed; (2) we used the new closed conformation structure obtained at pH 5.0 (PDB: 8X2H) to replace the old one; (3) the previous Fig. 8 (now Fig. 7) was also modified.

Reviewer #3

Suggestions

- *With these revisions and data to support the protective phenotype of CROPs across the LCTs and in maintaining virulence capacity for TcdA and TcdB, as well as TcsL, the work presented here represents an important advancement of knowledge of significance to the Clostridial field.*
- *I am willing to accept these modifications for publication. However, the authors may want to comment on how this data relates to TpeL which lacks a functional CROPs domain.*

Response: We very much thank the reviewer for supporting our study. We totally agree that TpeL is unique among LCTs as it has no CROPs domain. As suggested by the reviewer, we have added some comment on TpeL in the Discussion section. (Line 268-271)